

# Robust metabolic syndrome risk score based on triangular areal similarity

Hyunseok Shin[1], Simon Shim[2] and Sejong Oh[3]

[1] Department of Computer Science, Dankook University, Youngin, Gyeonggi, South Korea
[2] Department of Applied Data Science, San Jose State University, San Jose, CA, United States
[3] Department of Software Science, Dankook University, Youngin, Gyeonggi, South Korea

## ABSTRACT

One of the limitations of currently-used metabolic syndrome (MetS) risk calculations is that they often depend on sample characteristics. To address this, we introduced a novel sample-independent risk quantification method called 'triangular areal similarity' (TAS) that employs three-axis radar charts constructed from five MetS factors in order to assess the similarity between standard diagnostic thresholds and individual patient measurements. The method was evaluated using large datasets of Korean ($n$ = 72,332) and American ($n$ = 11,286) demographics further segmented by sex, age, and race. The risk score exhibited a strong positive correlation with the number of abnormal factors and was closely aligned with the current diagnostic paradigm. The proposed score demonstrated high diagnostic accuracy and robustness, surpassing previously reported risk scores. This method demonstrated superior performance and stability when tested on cross-national datasets. This novel sample-independent approach has the potential to enhance the precision of MetS risk prediction.

## INTRODUCTION

Metabolic syndrome (MetS) is a constellation of risk factors that directly affect atherosclerotic cardiovascular disease (CVD) (*Grundy et al., 2005*). These constellations are factors of interrelated metabolic origin that are strongly associated with insulin resistance and type 2 diabetes (*Grundy et al., 2005*; *Kahn et al., 2005*).

The criteria for the diagnosis of MetS were established by several specialized groups, such as the World Health Organization (WHO), the European Group for the Study of Insulin Resistance (EGIR), the National Cholesterol Education Program-Third Adult Treatment Panel (NCEP-ATP III), the American Association of Clinical Endocrinologists; American Heart Association-National Heart, Lung, and Blood Institute (AHA/NHLBI), and the International Diabetes Federation (IDF) (*Jeong et al., 2014*). Although the diagnostic criteria presented by each group vary slightly, MetS is commonly diagnosed on the basis of five risk factors: elevated waist circumference (WC), elevated fasting glucose (GL), elevated blood pressure (BP), elevated triglycerides (TG), and reduced high-density lipoprotein cholesterol (HDL). For each factor, abnormalities are determined using a specific threshold, and a diagnosis of MetS ensues when there are three or more abnormal factors present (*Stone, Bilek & Rosenbaum, 2005*).

Corresponding authors
Simon Shim, simon.shim@sjsu.edu
Sejong Oh, sejongoh@dankook.ac.kr

The MetS diagnosis based on thresholds has been widely used in clinical practice and epidemiological studies (*Grundy et al., 2005*). However, over the past 20 years, the limitations of categorical diagnosis have steadily increased (*Kahn et al., 2005*; *Eisenmann, 2008*; *Okosun et al., 2010*; *Jeong et al., 2014*; *Soldatovic et al., 2016*; *Wiley & Carrington, 2016*). MetS is a chronic disease that should be considered a progressive condition. However, because the current diagnostic criteria are dichotomous, information loss is inevitable. The current counting method does not sense nor reflect changes in risk factors. To overcome this limitation, various MetS risk scores with continuous values have been proposed (*Eisenmann, 2008*; *Okosun et al., 2010*; *Jeong et al., 2014*; *Soldatovic et al., 2016*; *Wiley & Carrington, 2016*).

The most common approach for deriving continuous risk scores is to use statistical techniques such as z-score, standardized residuals of linear regression, or principal component analysis (*Eisenmann, 2008*; *Khazdouz et al., 2021*). *Khazdouz et al. (2021)* collected and analyzed 1,113 studies related to continuous MetS (cMets) scores from 1980 to 2020 and ultimately selected 10 studies. However, most of these studies primarily employed the z-score approach, indicating that recent research on cMetS score calculation methods has not significantly diverged from established practices.

Among the key methodologies, *Okosun et al. (2010)* derived the cMets by: (1) converting the values of individual MetS factors into z-scores; (2) performing a linear regression depending on age, sex, and race; and (3) summing the standardized residuals obtained for each MetS factor. The diagnostic performance of the area under the receiver operating characteristic curve (AUC), recall, and specificity were 0.885, 0.831, and 0.833, respectively, with the population made up of American adults. The diagnostic threshold was set as the point at which the sum of recall and specificity was maximized, according to *Youden*'s *(1950)* index. However, cMetS is limited in that it is difficult to compare different populations because of dependency on the sample in the calculation process. In addition, it is difficult to interpret the meaning of cMetS intuitively because its range is not limited.

Another statistic-based score is the MetS severity score (MetSSS) proposed by *Wiley & Carrington (2016)*. MetSSS is derived by: (1) subtracting thresholds from individual MetS factor values, (2) zeroing values below 0, (3) converting these values into z-scores, (4) calculating the factor loading value of the MetS factors, (5) multiplying the factor loading value by the z-score, and (6) calculating the standardized distance. Although the diagnostic performance was not excellent among Europeans (with an AUC of 0.77, accuracy of 0.68, recall of 0.82, and specificity of 0.57), the sample-dependent properties were notably reduced by the adjustment method using the diagnostic threshold. However, MetSSS has similar limitations as cMetS, such as sample-dependent properties inherent in the calculation process and a lack of a clear range of values.

Another statistical approach is the siMS method, a simple technique for quantifying MetS introduced by *Soldatovic et al. (2016)*. The siMS score consists of a single sum of a line of formulas. This score is obtained by summing up each MetS component in the form of a linear regression. This simple method exhibited high diagnostic performance, with an AUC of 0.926. The sample-independent properties are an additional advantage of using MetS factor values without transformation. However, the population was relatively small

with 528 Serbians, and the correlation between the number of MetS factors and siMS was 0.745, indicating the need for additional verification and improvement.

Approaches other than statistics-based methods have also been proposed. Notably, a method exists for evaluating risk using the area of radar charts (*Jeong et al., 2014*), which are graphical representations designed to display multiple performance indicators concurrently in a circular format (*Jeong et al., 2014*; *Stafoggia et al., 2011*). Each indicator, standardized between 0 and 1, is positioned along the axes radiating from the center of the circle and connected to form a closed polygon (*Jeong et al., 2014*; *Saary, 2008*; *Stafoggia et al., 2011*). As a result, radar charts are valuable for visualizing multivariate data and provide an intuitive means of comparing overall performance (*Jeong et al., 2014*; *Saary, 2008*; *Peng et al., 2019*). Radar charts are widely utilized in business management, social science, and general engineering for performance measurement and risk assessment, and are being increasingly applied in healthcare, medicine, and biomedical engineering (*Jeong et al., 2014*; *Saary, 2008*; *Peng et al., 2019*).

*Jeong et al. (2014)* pioneered the use of radar charts for risk assessment in their groundbreaking study on MetS and introduced the areal similarity degree (ASD) method tailored for this purpose. ASD was derived by calculating the intersection area of the radar chart. Each MetS factor is represented by an axis on the radar chart. For each of the two adjacent axes, ASD quantifies the degree of overlap between a chart consisting of thresholds and the chart consisting of MetS factor values. The final score is derived by adding each resulting value to reflect the weight of each MetS factor. After evaluating 5,335 Koreans in various subgroups by sex and age, a strong correlation was observed between the number of risk factors and ASD in all groups. One advantage of ASD is that it ranges between 0 and 1, which allows for an intuitive interpretation. The closer it is to one, the closer the chart areas of MetS factors and thresholds. Another advantage of ASD is that weight reflects the importance of MetS factor.

However, ASD has three limitations. First, ASD is affected by the arrangement of the axes. The ASD method constructs a weighted radar chart with different central angles in proportion to the importance of MetS factors. In this design, the shape of the polygon depends on the arrangement of the axes and is asymmetric along the axis because of the different center angles. Second, ASD depends on the maximum and minimum values of the sample because it uses min–max scaling, which is vulnerable to outliers. Third, ASD exhibits sample-dependent properties. The weight of each MetS factor is determined by counting the number of specific cases in which the factor occurs within the sample. This count is then used to assign a measure of relative importance to each factor within the sample.

The cMetS risk score introduced thus far has one or more of the following limitations: (1) sample-dependent attributes, (2) uncertainty in the range or interpretation of values, (3) insufficient performance, and (4) variability under detailed conditions.

In this study, we introduce a robust risk score, the Robust MetS Risk Score (RMRS), which was designed to overcome these limitations. The 'Methods' section provides a detailed explanation for calculating RMRS with a specific focus on the innovative triangular areal similarity (TAS) method developed in this study. In the 'Results' section,

we objectively compare the robustness of the RMRS across sex, age, and race, using extensive datasets from Korean and American populations, alongside previously introduced risk scores and using objective measures. The 'Discussion' section critically examines the reasons for the superior performance of the proposed method and reflects on the broader significance of our study. The 'Conclusion' section summarizes the key findings and outlines potential directions for future studies.

## METHODS

This section presents the proposed TAS method, which creates an RMRS by integrating two triangular areas on the radar charts. This process is schematically illustrated in Fig. 1. The details of each block are presented in the following subsections.

### Defining risk factors

The risk factors and thresholds for MetS are based on the criteria proposed by the AHA/NHLBI. The AHA/NHLBI criteria are a revised version of the NECP-ATP III criteria, which have been widely used in both clinical practice and epidemiological studies (*Grundy et al., 2005*). Based on the NECP-ATP III (*Expert Panel on Detection, Evaluation, Treatment of High Blood Cholesterol in Adults, 2001*), the AHA/NHLBI clarified the definition of hypertension, lowered the fasting glucose threshold, and adjusted WC threshold to suit ethnicity (*Grundy et al., 2005*). Among Koreans, the WC threshold follows the criteria suggested by the Korean Society for Obesity (*Kim et al., 2020*). The five factors and thresholds for diagnosing MetS are presented in Table 1.

### Configuring radar chart

The TAS method begins by configuring the risk factors into a plurality of radar charts. Each risk factor is mapped onto each axis of the radar chart. In this process, we designed a radar chart with a consistent interpretation of the axis scale regardless of the sample dataset such that the area of the chart would not change according to the order of the axes.

#### Scale axes

The examination values for each risk factor with different scales must be adjusted to be between 0 and 1. Scale transformation sequentially performs normalization and sigmoid transformation. Normalization is performed as in Eq. (1) such that the threshold can be at 0, and 10% of the threshold is one unit size.

$$Z_{f_i} = \begin{cases} \left(x_{f_i} - d_{f_i}\right)/\left(0.1 * d_{f_i}\right) & if \ f_i \in \{GL, TG, \ WC\} \\ -\left(x_{f_i} - d_{f_i}\right)/\left(0.1 * d_{f_i}\right) & if \ f_i \in \{HDL\} \\ max\left(x_{f_{sbp}} - d_{f_{sbp}}, x_{f_{dbp}} - d_{f_{dbp}}\right)/\left(0.1 * d_{f_{bp}}\right) & if \ f_i \in \{BP\} \end{cases} \tag{1}$$

where $x_{f_i}$ is the examination value of risk factor $f_i$, and $d_{f_i}$ is the threshold of risk factor $f_i$. The HDL value is multiplied by −1 to match the interpretation of other factors. The opposite property applies: as HDL decreases, risk increases. For blood pressure, the larger of the two is selected when subtracting each threshold from the systolic and diastolic blood pressures: $max\left(x_{f_{sbp}} - d_{f_{sbp}}, \ x_{f_{dbp}} - d_{f_{dbp}}\right)$. The selected value is then adjusted so that the

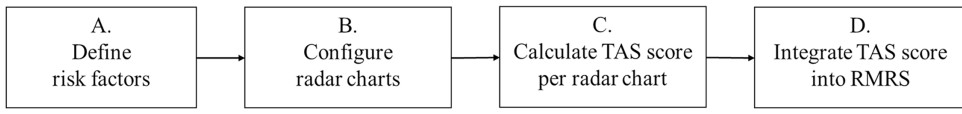

**Figure 1** **Main process of the triangular areal similarity (TAS) method.**

**Table 1** **Five risk factors of metabolic syndrome.**

| Risk factor (unit) | Thresholds |
| --- | --- |
| Fasting plasma glucose (**mg/dl**) | ≥100 |
| Blood pressure (**mmHg**) | *Systolic* ≥ 130 *or Diastolic* ≥ 85 |
| Triglycerides (**mg/dl**) | ≥150 |
| HDL-cholesterol (**mg/dl**) | *Male*: <40, *Female*: <50 |
| Waist circumference (**cm**) | American: *Male*: ≥102, *Female*: ≥88 Korean: *Male*: ≥90, *Female*: ≥85 |

size of one unit would be the same using $d_{f_{bp}} = d_{dbp} - d_{f_{sbp}}$, the difference between the two thresholds, rather than each threshold.

$Z_{f_i}$ is converted again using the Elliot sigmoid function, as shown in Eq. (2).

$$S_{f_i} = f(Z_{f_i}) = 0.5 * Z_i / (1 + |Z_{f_i}|) + 0.5. \tag{2}$$

The sigmoid transformation is robust to outliers and can map outlier values onto a standardized range of 0 to 1. Additionally, the sigmoid function exhibits continuous and monotonic properties with symmetry around (0, 0.5) (*Zou et al., 2023*). Furthermore, it possesses characteristics that make it suitable for modeling risk, including 1) an increase in risk with higher values, 2) rapid sensitivity to changes in risk near the threshold, and 3) the ability to model varying rates of change across intervals (*Hau, Amorim & Bergamin Filho, 1993*; *Zou et al., 2023*). Consequently, an axis consisting of $S_{f_i}$ has the following properties.

1) $S_{f_i}$ is the value of each of the five risk factors: WC, BP, HDL, GL, and TG.
2) 0.5 is the threshold for each MetS risk factor, as $f(Z_{f_i} = 0) = 0.5$
3) 0.25 is one unit size smaller than the threshold, as $f(Z_{f_i} = -1) = 0.25$
4) 0.75 is one unit size larger than the thresholds, as $f(Z_{f_i} = 1) = 0.75$
5) The axis is not linear.

### Combination of axes
Using three axes, we constructed a radar chart. Given the three axes, a, b, and c, two types of radar charts (in the order of a-b-c or a-c-b) were constructed with respect to the a-axis and the triangles represented by the two charts were congruent. Therefore, the three-axis radar chart can be used to calculate the area of a triangle without being affected by the order of the axes.

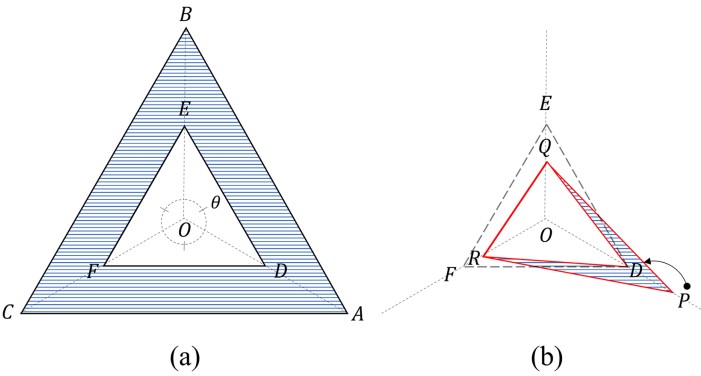

(a)                (b)

**Figure 2 Four triangles on a three-axis radar.** (A) Two criteria triangles: the critical triangle (ΔDEF) and the maximum triangle (ΔABC). Area (blue) beyond the critical triangle: C_β (B) Two measurement triangles: an internal triangle (ΔDQR) and an external triangle (ΔPQR). For the internal triangle configuration, the value P exceeding the threshold is limited to D (=0.5). Areas (blue) beyond the internal triangle: S_β. 

Next, we generated all possible combinations of the three-axis radar charts from the five MetS risk factors, resulting in a set of 10 basic graph units that were used to classify MetS into ten subtypes. These basic graph units are consistent with the diagnostic perspective, as the MetS diagnosis is based on the presence of at least three risk factors. Complementing each other, these graph units provide a more comprehensive understanding of MetS than a single three-axis radar chart, capturing information that cannot be captured by a single chart.

## Calculating TAS score per radar chart

The TAS is a scoring method that uses four triangles on a three-axis radar chart, as shown in Fig. 2.

### Four triangles

The four triangles are divided into two groups: criteria and measurement triangles.

The two criteria triangles refer to the threshold triangle (ΔDEF), to which each threshold (=0.5) is connected, and the maximum triangle (ΔABC), to which each maximum value (=1) is connected (Fig. 2).

The areas of the criteria triangles can be obtained using the sine law as follows:

$$S_{\Delta ABC} = \frac{1}{2} * sin\left(\frac{\pi}{3}\right) * \left(\overline{OA} * \overline{OB} + \overline{OB} * \overline{OC} + \overline{OC} * \overline{OA}\right), \text{where } \overline{OA} = \overline{OB} = \overline{OC} = 1 \quad (3)$$

$$S_{\Delta DEF} = \frac{1}{2} * sin\left(\frac{\pi}{3}\right) * \left(\overline{OD} * \overline{OE} + \overline{OE} * \overline{OF} + \overline{OF} * \overline{OD}\right), \text{ where } \overline{OD} = \overline{OE} = \overline{OF} = 0.5. \quad (4)$$

The two measurement triangles refer to the internal triangle (ΔDQR) connecting each value, with the maximum value limited to the threshold value, and the external triangle (ΔPQR) connecting the original values (Fig. 2). The original input values $X = [x_1, x_2, x_3]$

of the radar chart are converted into $X' = \left[x'_1, x'_2, x'_3\right]$ to construct an internal triangle using Eq. (5).

$$X' = [\min(x_1, 0.5), \min(x_2, 0.5), \min(x_3, 0.5)]. \tag{5}$$

The area of the measurement triangles can be obtained using the sine law as follows:

$$S_x := S_{\Delta PQR} = \frac{1}{2} * sin\left(\frac{\pi}{3}\right) * (x_1 x_2 + x_2 x_3 + x_3 x_1) \tag{6}$$

$$S_{x'} := S_{\Delta DQR} = \frac{1}{2} * sin\left(\frac{\pi}{3}\right) * \left(x'_1 x'_2 + x'_2 x'_3 + x'_3 x'_1\right). \tag{7}$$

### TAS score

The TAS score is calculated by recombining the four previously obtained triangles and focusing on the following attributes:

1) $S_{\Delta ABC}$ and $S_{\Delta DEF}$ are constants.
2) Let $C_\alpha := S_{\Delta DEF}$ and $C_\beta := S_{\Delta ABC} - S_{\Delta DEF}$, then $C_\beta = 3 * C_\alpha$; $C_\alpha$ is an area-based diagnostic criterion for these three risk factors; $C_\beta$ is the maximum value of 'the area-based severity,' which is the maximum excess allowance from $C_\alpha$ (Fig. 2A).
3) Let $S_\beta := S_x - S_{x'}$, then $S_x$ can be decomposed into $S_{x'}$ and $S_\beta$ (Fig. 2B). The range of $S_\beta$ is between 0 and $C_\beta$.

The idea of TAS is as follows:

1) $S_{x'}$ and $S_\beta$ contain different types of information. $S_{x'}$ contains a degree close to the diagnostic threshold of MetS, and $S_\beta$ contains a serious degree that is not reflected in the diagnosis.
2) The effects of $S_\beta$ differ before and after MetS onset.
3) These differences are adjusted by reflecting $C_\alpha$ and $C_\beta$.

Based on these properties and ideas, the formula for the TAS score is defined as

$$TAS(x_1, x_2, x_3) = \frac{1}{2} \left\{ closeness\left(S_{x'}, S_\beta\right) + severity\left(S_{x'}, S_\beta\right) \right\}$$

$$= \frac{X}{0.75 + X - X'} + I(S_{x'}) * \frac{4}{9}(X - X')$$

*where*:

$$closeness\left(S_{x'}, S_\beta\right) = \frac{S_{x'} + S_\beta}{C_\alpha + S_\beta}, \; severity\left(S_{x'}, \; S_\beta\right) = I(S_{x'}) * \frac{S_\beta}{C_\beta},$$

$$I(S_{x'}) := if \; S_{x'} = C_\alpha \; then \; 1 \; else \; 0,$$

$$X = x_1 x_2 + x_2 x_3 + x_3 x_1, \; X' = x'_1 x'_2 + x'_2 x'_3 + x'_3 x'_1 \tag{8}$$

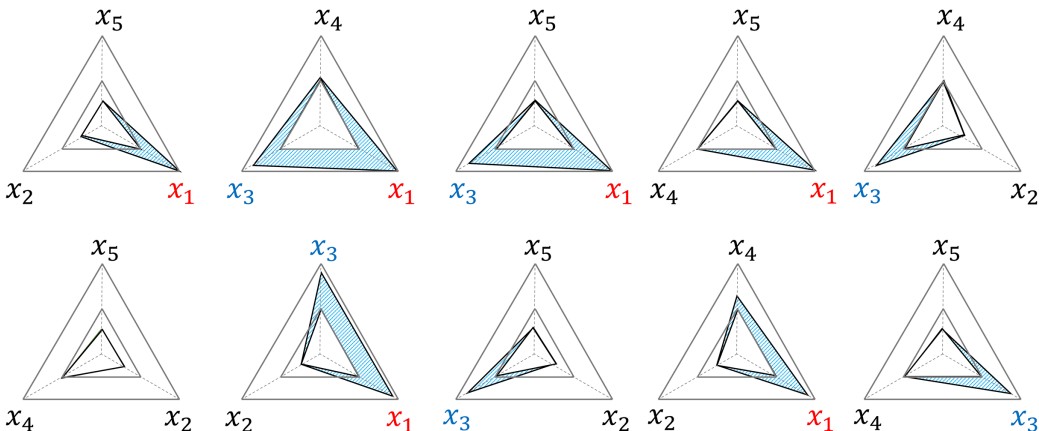

**Figure 3 Examples of structural weighted effects.** $x_1$ and $x_3$ are factors that exceed the threshold, and $S_\beta$ (blue area) is weighted in six out of 10 TAS scores, respectively.

Properties:

1) The TAS score is the average of the closeness and severity functions and has a value between 0 and 1. Closeness is defined as the degree of proximity to MetS.

2) Before the onset of MetS ($S_{x'} < C_\alpha$), $S_\beta$ only affects the closeness function and not the severity function. Based on the ratio of $S_{x'}$ to the maximum value ($= C_\alpha$) of $S_{x'}$, the closeness function is modified to reflect severity $S_\beta$ and is in the range of 0 and 1.

3) After the onset of MetS ($S_{x'} = C_\alpha$), $S_\beta$ affects only the severity function and not the closeness function. The severity function is the ratio of $S_\beta$ to the maximum value ($= C_\beta$) of $S_\beta$, which is in the range of 0 to 1.

## Integrating TAS score into RMRS
### RMRS
RMRS is defined as Eq. (9). The RMRS is between 0 and 1 because it is the square root of the mean of the TAS scores which range from 0 to 1.

$$RMRS = \sqrt{\frac{1}{10}\sum_{i=1}^{10} TAS\ score_i}. \tag{9}$$

The RMRS has a structural characteristic that adds to the impact of the risk factors beyond the threshold. The example in Fig. 3 delineates the area (blue) where $S_\beta$ occurs in each radar chart because $x_1$ and $x_3$ exceed the threshold among the five risk factors; $x_1$ and $x_3$ add to this effect by generating $S_\beta$ in a total of six radar charts. In addition, the three radar charts (second and third graphs in the upper row and second graph in the lower row) further reflect the interaction effects of $x_1$ and $x_3$.

### Diagnostic threshold
A diagnostic threshold was established so that RMRS could also be used to diagnose MetS. Although the current diagnostic criteria for MetS require the presence of at least three risk

factors, the RMRS is a continuous value that can vary within a certain range, even when the three factors are equal. To address this, we calculated the distribution ranges of MetS risk factors for both two- and three-risk factor scenarios and set the threshold for MetS diagnosis at the center of the interval where the two ranges overlapped. The overlapping range was defined as the range from the minimum RMRS score (three risk factors) to the maximum RMRS score (two risk factors). The calculation was performed using two inputs for the risk factors, constructed regardless of order, namely, [0.5, 0.5, 0.5, 0, 0] and [1, 1, 0.5-$\epsilon$, 0.5-$\epsilon$, 0.5-$\epsilon$, 0.5], where $\epsilon$ is an arbitrarily small value close to zero. Based on the calculation results of 0.387 and 0.707, a threshold value of 0.547 was determined as the center of the overlapping range. The threshold is based only on the structural properties of the RMRS and is consistently applicable to all samples.

## Dataset and software

This study evaluates the performance of the Korean Genome and Epidemiology Study Health Examination (KoGES_HEXA) and National Health and Nutrition Examination Survey (NHANES) datasets. For more information on each dataset, the reader is referred to *Kim, Han & KoGES group (2017)* and the *National Center for Health Statistics (2022)*. The Institutional Review Board (IRB) of Dankook University approved the study protocol and waived the requirement for obtaining informed consent from the participants (DKU 2021-06-008). The analysis was performed using R software, version 4.2.2 (*R Core Team, 2022*).

KoGES_HEXA is a city-based Korean cohort dataset collected by the Korea Disease Control and Prevention Agency (KDCPA) since 2004 for chronic disease research (*Kim, Han & KoGES group, 2017*). KoGES_HEXA was released and includes the baseline survey dataset conducted between 2004–2013 and the first follow-up survey dataset conducted between 2012–2016 (*Korea National Institute of Health, 2022*). We retrieved these pseudonymized datasets with the approval of KDCPA and finally used a baseline survey dataset with a relatively large sample size ($n$ = 173,209).

The NHANES dataset was collected by the Centers for Disease Control and Prevention (CDC) to assess the health and nutritional status of adults and children in the United States. Since 1999, NHANES has been continuously conducted, with the publicly available data released every 2 years. We collected the public data from 2003 to 2020 from the CDC website and constructed 86,618 samples (*National Center for Health Statistics, 2022*).

Both datasets contained anthropometric data, disease history, drug intake, and the necessary information for the diagnosis of MetS. Using this additional information, participants from the 72,332 subjects selected from the KoGES_HEXA and 11,286 subjects from the NHANES (Table 2) with conditions that could affect the MetS diagnosis were excluded from each dataset. These conditions included: (1) hypertension, diabetes, hyperlipidemia, stroke, fatty liver, angina pectoris, thyroid disease, and cancer; (2) taking medication for hypertension or hyperlipidemia; (3) pregnant women; (4) over 80 years of age; and (5) missing values and outliers regarding diet, blood tests, and anthropometric measurements.

**Table 2  Characteristics of study subjects.**

| Dataset | KoGES_HEXA ($n$ = 72,332) | | NHANES ($n$ = 11,286) | |
|---|---|---|---|---|
| Sex | Male | Female | Male | Female |
| $n$ | 22,215 | 50,117 | 5,948 | 5,338 |
| MetS (%) | 18.7 | 10.5 | 19.0 | 17.2 |
| Age (years) | 51.9 ± 8.6 | 50.5 ± 7.6 | 34.3 ± 15 | 34.4 ± 14.9 |
| Range | 40–79 | 40–79 | 16–79 | 16–79 |
| 20< | – | – | 21.9% | 21.1% |
| 20–39 | – | – | 44.0% | 44.2% |
| 40–59 | 78.4% | 86.0% | 26.3% | 27.0% |
| 60≥ | 21.6% | 14.0% | 7.77% | 7.66% |
| GL | 94.8 ± 17.5 | 89.9 ± 13 | 100.2 ± 19.3 | 95.3 ± 15.1 |
| WC | 84.6 ± 7.4 | 77.3 ± 7.8 | 94.2 ± 15.5 | 91 ± 16.1 |
| HDL | 50.4 ± 12.1 | 57.3 ± 12.9 | 49.6 ± 13.5 | 58.3 ± 15.3 |
| TG | 145.3 ± 105.6 | 104.4 ± 66.7 | 122.7 ± 116.2 | 94.4 ± 70 |
| SBP | 124 ± 14.3 | 118.3 ± 14.7 | 119.6 ± 13 | 112.5 ± 13.9 |
| DBP | 77.9 ± 9.8 | 73.4 ± 9.6 | 69.3 ± 12.1 | 67.1 ± 10.5 |

**Note:**

GL, Fasting glucose (ml/dl); WC, Waist circumference (cm); HDL, HDL-cholesterol (ml/dl); TG, Triglycerides (ml/dl); SBP, Systolic blood pressure (mm Hg); DBP, Diastolic blood pressure (mm Hg).

## RESULTS

### The number of risk factors and the risk score

Since current MetS diagnostic criteria are based on the number of factors exceeding the threshold, *i.e.*, the number of risk factors, RMRS analyzed the relevance of the number of these risk factors. The distribution of the number of risk factors is shown in Fig. 4A; the ratio decreased as the number increased. This trend appears in both datasets, with similar ratios (%) of each number to the whole: 36.9, 31.0, 19.2, 9.4, 3.1, and 0.6 for KoGES_HEXA, and 30.7, 29.8, 21.3, 12.1, 4.8, and 1.2 for NHANES (Table 3).

Figure 4B shows the distribution of RMRS, with each bar in Fig. 4A converted into a histogram. Figure 4B shows how the existing discrete space (number of risk factors) expands into a continuous space. As the measurement space expands with the continuous risk score, an overlapping interval occurs between each histogram corresponding to the number of risk factors.

The range of the risk score column in Table 3 shows the theoretically possible range for the number of risk factors. For the two empirical datasets, the ranges were narrower than the theoretical range. In addition, as the number of risk factors increased, the minimum, maximum, and average values of the range increased proportionally. Pearson's correlation analysis showed a statistically significant positive correlation ($p < 0.05$) between the number of risk factors and the average, maximum, and minimum values of the risk score: 0.9997, 0.9946, and 0.9806 for KoGES_HEXA and 0.9993, 0.9937, and 0.9611 for NHANES, respectively.

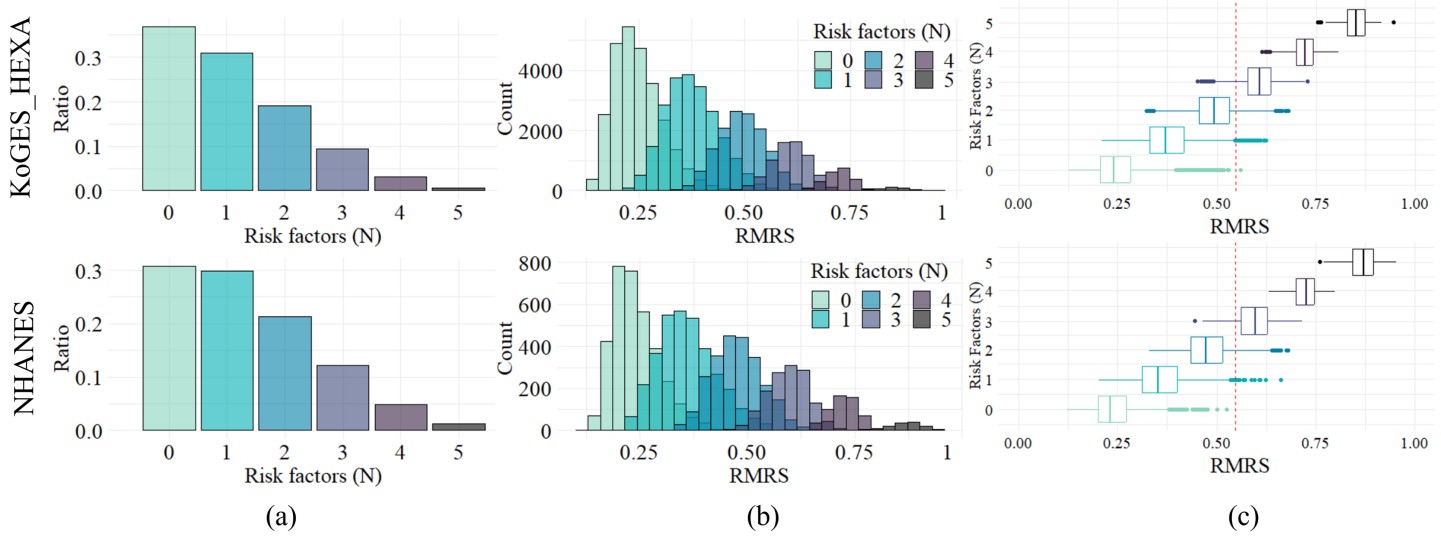

**Figure 4 Distribution and relationship of number of risk factors and risk indices.** (A) Composition ratio by number of risk factors. (B) Distribution of RMRS. (C) The boxplot of RMRS by the number of risk factors. The vertical line (red dotted) is 0.574, the proposed threshold for MetS diagnosis.

**Table 3 Distribution of risk score by number of MetS risk factors.**

| Risk factors (*N*) | Range of risk score | KoGES_HEXA | | NHANES | |
|---|---|---|---|---|---|
| | | Mean ± sd (Range) | % | Mean ± sd (Range) | % |
| 0 | 0–0.707 | 0.248 ± 0.057 (0.127–0.560) | 36.9 | 0.240 ± 0.055 (0.123–0.525) | 30.7 |
| 1 | 0–0.707 | 0.375 ± 0.062 (0.210–0.624) | 31.0 | 0.359 ± 0.067 (0.203–0.661) | 29.8 |
| 2 | 0.224–0.707 | 0.494 ± 0.055 (0.322–0.681) | 19.2 | 0.476 ± 0.058 (0.330–0.681) | 21.3 |
| 3 | 0.387–0.742 | 0.606 ± 0.044 (0.451–0.729) | 9.4 | 0.595 ± 0.046 (0.445–0.716) | 12.1 |
| 4 | 0.548–0.837 | 0.720 ± 0.033 (0.613–0.807) | 3.1 | 0.723 ± 0.033 (0.632–0.797) | 4.8 |
| 5 | 0.707–1 | 0.850 ± 0.033 (0.753–0.945) | 0.6 | 0.866 ± 0.040 (0.760–0.953) | 1.2 |

Figure 4C is a display of Fig. 4B in boxplot format where the distribution characteristics of the RMRS according to the number of risk factors are clearly observed. For each number of risk factors, the RMRS scores corresponding to the 2nd and 3rd quartiles did not overlap. However, the RMRS corresponding to the fourth quartile overlapped the RMRS with more risk factors. Therefore, even with fewer than three risk factors, some cases fell into the range of MetS diagnoses based on RMRS (refer to the example provided in Table S1).

## Comparison between methods

The overall performance of RMRS was evaluated by constructing the KoGES_HEXA and NHANES datasets as subgroups of 28 cases according to sex, age group, and race. The 28 subgroups are shown in Table S2 and consist of sizes ranging from 409 to 72,332. In

Table 4 Performance comparison between MetS risk scores.

| | cMetS (2008) | ASD (2014) | siMS (2016) | MetSSS (2016) | RMRS (Proposed) |
|---|---|---|---|---|---|
| Correlation | 0.749 ± 0.036 | 0.805 ± 0.015 | 0.728 ± 0.045 | 0.626 ± 0.034 | **0.927** ± 0.009 |
| Adjusted R square | 0.562 ± 0.053 | 0.649 ± 0.024 | 0.532 ± 0.065 | 0.392 ± 0.042 | **0.859** ± 0.016 |
| AUC | 0.929 ± 0.025 | 0.953 ± 0.015 | 0.924 ± 0.021 | 0.878 ± 0.029 | **0.989** ± 0.004 |
| Accuracy | 0.851 ± 0.029 | 0.917 ± 0.029 | 0.843 ± 0.025 | 0.752 ± 0.031 | **0.950** ± 0.015 |
| Recall | 0.878 ± 0.04 | 0.670 ± 0.092 | 0.874 ± 0.047 | **0.900** ± 0.046 | 0.888 ± 0.051 |
| Specificity | 0.847 ± 0.032 | **0.966** ± 0.030 | 0.837 ± 0.034 | 0.721 ± 0.047 | 0.962 ± 0.02 |
| Thresholds | 1.397 ± 0.522 | 0.943 ± 0.020 | 2.005 ± 0.083 | 1.253 ± 0.260 | **0.547** (fixation) |

Note:
The average performance (mean ± standard deviation) for different subgroups (total 28 cases) by dataset, sex, age group, and race. The best performance is marked in bold.

addition, an objective performance comparison of the RMRS was attempted by comparing the four previously proposed scoring methods: cMetS (*Okosun et al., 2010*), ASD (*Jeong et al., 2014*), siMS (*Soldatovic et al., 2016*), and MetSSS (*Wiley & Carrington, 2016*), under the same conditions. Their performance was evaluated by focusing on the correlation between the number of risk factors, risk scores, and metrics related to MetS discrimination, such as AUC, accuracy, recall, and specificity. *Youden*'s *(1950)* index was used to determine the optimal thresholds for these methods, with the exception of the ASD method, with guidance on the threshold setting required for MetS diagnosis. The RMRS was the only method that used a fixed threshold (0.547), regardless of the subgroup. In addition, cMetS, ASD, and MetSSS scores were calculated for each subgroup considering sample-dependent characteristics.

Table 4 summarizes the evaluation results for each score. RMRS outperformed the other scores in most evaluation metrics. First, the correlation coefficient of RMRS was 0.927, which was significantly higher than that of the other scores. This result supports the strong correlation between RMRS and the number of risk factors, which is the current diagnostic criterion. Second, the AUC and accuracy were the highest at 0.989 and 0.950, respectively. This result confirms that RMRS has high discrimination and accuracy, even for MetS diagnosis. The MetSSS and ASD showed the highest recall (0.900) and specificity (0.966), respectively. Although the RMRS recall (0.888) and specificity (0.962) were not the highest, the difference in performance was not significant. Notably, RMRS results were obtained using a fixed threshold. Third, we identified the robustness of the RMRS in various subgroups according to nation, sex, age group, and race. The deviations in the evaluation results within the subgroups (Table 4 and Fig. S1) show that RMRS is concentrated in a narrower range than the other scores.

## DISCUSSION

The RMRS was proposed based on the TAS method to overcome the limitations of existing methods. Our method shares the underlying idea of ASD in calculating the overlay area of radar charts, but reflects robust and sample-independent properties. ASD constructs a radar chart using the center angle as the weight, resulting in asymmetric polygons along the axis. Although this asymmetry reflects the importance of the MetS factors, it also leads

to deviations in the risk score. We addressed this problem by reconstructing the five-axis radar chart of ASD in the form of 10 three-axis radar charts. Consequently, this difference also reflects the weights. The ASD reflects global weights as the ratio of each anomaly to the overall anomaly of the sample, whereas RMRS reflects instance-level weights using the method shown in Fig. 3.

The RMRS structurally reflects the effect of the interaction between MetS factors. Although the TAS score calculation was derived from the triangular area, the equation can be interpreted as the sum of the interaction effects between each factor. The reconstruction of TAS Formula (8) from the interaction perspective is defined in Eq. (10): Input values $x_1$, $x_2$, and $x_3$ are replaced with $0.5 + \alpha_1$, $0.5 + \alpha_2$, and $0.5 + \alpha_3$, respectively, based on the threshold of 0.5.

$$TAS(x_1, x_2, x_3) = TAS(0.5 + \alpha_1, 0.5 + \alpha_2, 0.5 + \alpha_3) = \frac{I_T + I_P + I_N}{I_T + I_P} + I * \left(\frac{4}{9}\right) * I_p,$$

where $I_T = 0.75$, $-0.5 < \alpha_1, \alpha_2, \alpha_3 < 0.5$, $I :=$ if $\alpha_1, \alpha_2, \alpha_3 \geq 0$ then 1 else 0

case 1 : $\alpha_1, \alpha_2, \alpha_3 < 0$,       $I_P = 0$,    $I_N = \alpha_1 + \alpha_2 + \alpha_3 + \alpha_1\alpha_2 + \alpha_2\alpha_3 + \alpha_3\alpha_1$

case 2 : $\alpha_1 \geq 0$, $\alpha_2, \alpha_3 < 0$,    $I_P = \alpha_1 + \alpha_1\alpha_2 + \alpha_3\alpha_1$,   $I_N = \alpha_2 + \alpha_3 + \alpha_2\alpha_3$

case 3 : $\alpha_1, \alpha_2 \geq 0, \alpha_3 < 0$,    $I_P = \alpha_1 + \alpha_2 + \alpha_1\alpha_2 + \alpha_2\alpha_3 + \alpha_3\alpha_1$,   $I_N = \alpha_3$

case 4 : $\alpha_1, \alpha_2, \alpha_3 \geq 0$,       $I_P = \alpha_1 + \alpha_2 + \alpha_3 + \alpha_1\alpha_2 + \alpha_2\alpha_3 + \alpha_3\alpha_1$,   $I_N = 0$.    (10)

where $I_P$ is the interaction caused by factors $\alpha_i \geq 0$, and $I_N$ is the interaction caused by factors $\alpha_i < 0$. $I_T$ is an interaction that occurs based on the threshold value (0.5) of each factor and is a reference interaction. In Cases 1–3, MetS was not diagnosed based on these three factors. In this case, only the left-hand side of Eq. (10) remains. Because $I_T$ and $I_p$ are common factors in the numerator and denominator, respectively, $I_N$ is eventually reflected as a penalty. In other words, even if the effect exceeding the threshold is large, if the effect below the threshold is also large, the two effects of TAS are offset. In contrast, Case 4 exceeds all thresholds, reflecting the interaction effect of all factors without a penalty.

Finally, the contribution of this study can be summarized as follows:

**Universal risk score without sample-dependence:** By eschewing sample-dependent approaches such as the z-score, PCA loading values for MetSSS and cMetS, and sample-based weights for ASD, a universal risk score can be established by RMRS. This is achieved by relying solely on the threshold of the MetS factor, thereby ensuring independence from sample variation.

**Fixed threshold stability:** RMRS introduces stability through a fixed threshold derived from the structural characteristics that are unaffected by sample variations. This stability is demonstrated across diverse sample groups including different countries, age groups, sexes, and races.

**Objective performance comparison with large datasets:** The risk score, originally introduced in our previous study (*Shin et al., 2021*), was confined to Korean data. However, our current study expanded the dataset to include American data and conducted a comprehensive comparative analysis with four other methods, thereby establishing the superior performance of our approach. This study pioneers cross-country performance

comparisons of existing risk scores by leveraging extensive Korean and American datasets collected by their respective governments.

**Clinical implications:** While recognizing the enduring value of dichotomous MetS scores in clinical practice, this study acknowledges the limitations in identifying early-stage abnormalities (*Khazdouz et al., 2021*). The proposed cMetS scores offer enhanced reliability in predicting MetS risk compared to traditional criteria. Given the global prevalence of MetS, our risk scores have substantial utility in preventive medicine worldwide.

## CONCLUSION

This study addressed the limitations of the existing dichotomous diagnostic method for MetS and previously proposed calculations for a cMetS score. We introduced the robust MetS risk score RMRS based on the TAS method with a sample-independent three-axis radar chart, further verifying that RMRS exhibits consistently superior performance with universal properties.

Further research and verification of this scaling method are required. After setting 10% of the threshold to one unit using Eq. (1), the 10% interval around the threshold was nonlinearly expanded using Eq. (2). The threshold setting of 10% should be verified clinically to determine its appropriateness. Whether it is reasonable to assume equal weighting for all MetS factors at 10% and whether it would be more appropriate to adjust this interval individually also require further exploration.

### Funding

This work was supported by the High-Potential Individuals Global Training Program from the Ministry of Science, ICT (MSIT), Korea (No. 2021-0-01531), the development of AI ophthalmologic diagnosis and smart treatment platform based on big data from the Institute for Information & Communications Technology Planning & Evaluation (IITP) (No. 2018-0-00242), and the data fabric technology to support logical data integration and compound analysis of distributed data from IITP (No. RS-2023-00222191). There was no additional external funding received for this study. The funders had no role in study design, data collection and analysis, decision to publish, or preparation of the manuscript.

### Grant Disclosures

The following grant information was disclosed by the authors:
High-Potential Individuals Global Training Program from the Ministry of Science, ICT (MSIT), Korea: 2021-0-01531.
Institute for Information & Communications Technology Planning & Evaluation (IITP): 2018–0-00242.
Data fabric technology to support logical data integration and compound analysis of distributed data from IITP: RS-2023-00222191.

## Competing Interests

The authors declare that they have no competing interests.

## Author Contributions

- Hyunseok Shin conceived and designed the experiments, performed the experiments, analyzed the data, performed the computation work, prepared figures and/or tables, authored or reviewed drafts of the article, and approved the final draft.
- Simon Shim conceived and designed the experiments, authored or reviewed drafts of the article, and approved the final draft.
- Sejong Oh conceived and designed the experiments, performed the experiments, authored or reviewed drafts of the article, and approved the final draft.

## Ethics

The following information was supplied relating to ethical approvals (*i.e.*, approving body and any reference numbers):

The study protocol was approved by the Institutional Review Board (IRB) of Dankook University. The IRB of Dankook University waived the requirement for obtaining informed consent from participants due to the use of non-identifiable data collected from public institutions (Approval number: DKU 2021-06-008).

## Data Availability

The NHANES data is publicly available on the U.S. Centers for Disease Control and Prevention website https://www.cdc.gov/nchs/nhanes/about_nhanes.htm.

Researchers must contact the Korea Disease Control and Prevention Agency's Institutional Data Access/Ethics Committee through the division of Population Health Research's website https://nih.go.kr/ko/main/contents.do?menuNo=300566 to obtain KoGES_HEXA data. Access to KoGES_HEXA data is restricted to researchers who meet the eligibility requirements for data access.

The experimental code is available at GitHub and Zenodo:

- https://github.com/shinhseok/triangular-areal-similarity.git
- Shin, H. (2023). Triangular areal similarity. Zenodo. https://doi.org/10.5281/zenodo.10599442.

## Supplemental Information

Supplemental information for this article can be found online at http://dx.doi.org/10.7717/peerj-cs.2015#supplemental-information.

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
