# Peer review of "Robust metabolic syndrome risk score based on triangular areal similarity"

_PeerJ Computer Science, doi:10.7717/peerj-cs.2015_

## Round 0.1 · original submission · Major Revisions

Dear authors,

You are advised to critically respond to all comments point by point when preparing a new version of the manuscript and while preparing for the rebuttal letter. Please address all the comments/suggestions provided by the reviewers.

Kind regards,
PCoelho

**Language Note:** The review process has identified that the English language must be improved. PeerJ can provide language editing services - please contact us at copyediting@peerj.com for pricing (be sure to provide your manuscript number and title). Alternatively, you should make your own arrangements to improve the language quality and provide details in your response letter. – PeerJ Staff

Reviewer 1 ·

Basic reporting

The article is well written in English, clear and technical. However, I found a few flaws in the punctuation: on lines 189, 376 and 410, the period is missing; on line 407, "At" is missing a space.

The introduction needs to be improved. The following changes should be considered:

1. I suggest making a comparison with other works by the same author and highlighting the contributions of this paper in relation to those papers:

Shin, Hyunseok, et al. "Robust metabolic syndrome risk index based on triangular areal similarity." 2021 IEEE International Conference on Bioinformatics and Biomedicine (BIBM). IEEE, 2021.

Shin, Hyunseok, et al. "Prediction of Metabolic Syndrome based on Non-invasive Measurement Features for Chronic Disease Management." Proceedings of the 2022 8th International Conference on Computer Technology Applications. 2022.

Shin, Hyunseok, Simon Shim, and Sejong Oh. "Machine learning-based predictive model for prevention of metabolic syndrome." Plos one 18.6 (2023): e0286635.

2. Before going into the methods, the author should outline the content that will be presented in the rest of the paper.

The paper does not include conclusions in its structure. This should be corrected.

Experimental design

In the experimental design I suggest that the radar chart method be more detailed, indicating when it is good to use this method, its basic definitions, with some bibliographic reference on this approach, as well as the indication of other papers that use this methodology.

I suggest citing some references for the sigmoid transformation method.

Validity of the findings

The paper does not present conclusions. This should be corrected.

Additional comments

The article presents a new method for quantifying risk for metabolic syndrome, independent of the sample, called "triangular area similarity". In my opinion, it is an interesting and innovative method and is eligible for publication once the flaws pointed out by this reviewer have been corrected.

Cite this review as

Reviewer 2 ·

Basic reporting

The present manuscript entitled “Robust metabolic syndrome risk score based on 2 triangular areal similarity” has scientific potential for the future researcher and clinicians. Authors must rewrite the introduction to include more recent study which will give more strength for this manuscript. Authors have to focus on the clinical relevance and novelty of the current manuscript. There are some typo and grammatical errors in the text. Authors has to rewrite the conclusion which will give scientific direction to the future researcher and clinicians in the current field metabolic syndrome

Experimental design

Experimental design is appropriate of the current manuscript.

Validity of the findings

The data of the present study is looking fine for me.

Additional comments

Authors has to focus on the clinical relevance of the current manuscript.

Cite this review as

---

## Round 0.2 · accepted · Accept

Dear authors, we are pleased to verify that you meet the reviewer's valuable feedback to improve your research.

Thank you for considering PeerJ Computer Science and submitting your work.

Reviewer 1 ·

Basic reporting

This reviewer is convinced of the authors' justifications and corrections. In my opinion, the paper can be accepted.

Experimental design

no comment

Validity of the findings

no comment

Additional comments

no comment

Cite this review as